# Immunogenic and Protective Properties of mRNA Vaccine Encoding Hemagglutinin of Avian Influenza A/H5N8 Virus, Delivered by Lipid Nanoparticles and Needle-Free Jet Injection

**DOI:** 10.3390/vaccines13080883

**Published:** 2025-08-21

**Authors:** Vladimir A. Yakovlev, Victoria R. Litvinova, Nadezhda B. Rudometova, Mariya B. Borgoyakova, Elena V. Tigeeva, Ekaterina V. Starostina, Ksenia I. Ivanova, Andrei S. Gudymo, Natalia V. Danilchenko, Olga N. Perfilyeva, Kristina P. Makarova, Danil I. Vahitov, Boris N. Zaitsev, Elena V. Dmitrienko, Sergey V. Sharabrin, Svetlana I. Krasnikova, Lyubov A. Kisakova, Denis N. Kisakov, Tatiana N. Ilyicheva, Vasiliy Yu. Marchenko, Larisa I. Karpenko, Andrey P. Rudometov, Alexander A. Ilyichev

**Affiliations:** 1Federal Budgetary Research Institution State Research Center of Virology and Biotechnology «Vector», Rospotrebnadzor, 630559 Koltsovo, Russia; viktoriya_litvinova_1999@mail.ru (V.R.L.); andreeva_nb@vector.nsc.ru (N.B.R.); borgoyakova_mb@vector.nsc.ru (M.B.B.); tigeeva_ev@vector.nsc.ru (E.V.T.); starostina_ev@vector.nsc.ru (E.V.S.); ivanova_ki@vector.nsc.ru (K.I.I.); gudymo_as@vector.nsc.ru (A.S.G.); danilchenko_nv@vector.nsc.ru (N.V.D.); perfileva_on@vector.nsc.ru (O.N.P.); makarova_kp@vector.nsc.ru (K.P.M.); vahitov_di@vector.nsc.ru (D.I.V.); zaitsev@vector.nsc.ru (B.N.Z.); sharabrin_sv@vector.nsc.ru (S.V.S.); krasnikova_si@vector.nsc.ru (S.I.K.); orlova_la@vector.nsc.ru (L.A.K.); def_2003@mail.ru (D.N.K.); ilicheva_tn@vector.nsc.ru (T.N.I.); andrei692@mail.ru (A.P.R.); ilyichev@vector.nsc.ru (A.A.I.); 2Institute of Chemical Biology and Fundamental Medicine, Siberian Branch, Russian Academy of Sciences, 630090 Novosibirsk, Russia; elenad@niboch.nsc.ru

**Keywords:** mRNA vaccines, lipid nanoparticles, LNPs, needle-free jet injection, mRNA delivery, hemagglutinin, highly pathogenic avian influenza virus, immune response, H5Nx

## Abstract

**Background/Objectives:** The development of a vaccine against highly pathogenic avian influenza viruses subtype A/H5 is an urgent task due to concerns about its pandemic potential. **Methods:** In this study, we have developed an experimental mRNA vaccine, mRNA-H5, encoding a modified hemagglutinin trimer of influenza virus A/turkey/Stavropol/320-01/2020 (H5N8). BALB/c mice were immunized with the mRNA-H5 vaccine using lipid nanoparticles (LNPs) and needle-free jet injection (JI). Subsequently, the immune response to vaccine was assessed using ELISA, microneutralization assay, and ICS methods, and a challenge study was conducted. **Results:** mRNA-H5 was shown to effectively stimulate specific humoral and T-cell immune responses. Moreover, mRNA-H5 delivered by LNPs and JI provided 100% protection of immunized mice against lethal challenge with homologous and heterologous strains of avian influenza virus (A/Astrakhan/3212/2020 (H5N8) and A/chicken/Magadan/14-7V/2022 (H5N1), respectively). **Conclusions:** The present results indicate that JI can be considered as an alternative to LNPs for mRNA delivery, and according to the literature, JI is safer than delivery using LNP. mRNA-H5 has potential as a vaccine against infection with highly pathogenic avian influenza A/H5 viruses with pandemic potential.

## 1. Introduction

As of 5 January 2024, highly pathogenic avian influenza (HPAI) A/H5 viruses of clade 2.3.4.4b have caused the culling of more than 541 million birds to prevent the spread of the disease, causing serious economic damage to the poultry industry [1]. But these virus subtypes pose a threat not only to birds but also to human health [2,3,4]. As of 20 June 2025, 976 cases of A/H5N1 infection in humans have been reported, of which 470 resulted in death [5]. HPAI A/H5 viruses have also been reported to infect a variety of wild and domestic mammals, including cattle, badgers, black bears, bobcats, coyotes, ferrets, cats, foxes, leopards, opossums, pigs, raccoons, skunks, sea lions, and wild otters [6]. Further circulation of HPAI A/H5 viruses among birds and mammals poses a threat of emergence of a new HPAI virus capable of human-to-human transmission and raises concerns about a possible pandemic caused by these influenza virus subtypes [7]. In response to these concerns, WHO calls for increased efforts to develop a vaccine against infection with HPAI viruses.

mRNA vaccines [8,9,10] have several properties that distinguish them from other types of influenza vaccines [11,12,13]. The mRNA vaccine platform has rapid, cost-effective, and scalable production in which the target immunogen can be rapidly replaced; it is particularly relevant due to the high variability of the influenza virus. mRNA vaccines do not elicit an immune response against themselves and, therefore, can be administered repeatedly. An important advantage is also the effective activation of both humoral and cellular virus-specific immune responses [14,15].

Lipid nanoparticles (LNPs) effectively protect exogenous mRNA from degradation by extracellular nucleases and are currently the most widely used system for delivering mRNA into cells [16,17,18,19,20]. For researchers around the world, the standard of LNP production is the composition developed by Moderna and Pfizer for COVID-19 vaccines [21,22]. The LNPs in these vaccines consist of four lipid components that ensure efficient encapsulation and release of mRNA, a long period of circulation in the body, and stability of the complex [23]. An additional advantage of LNPs is their adjuvant effect that enhances the immune response to the vaccine [24].

Alternative delivery methods have recently attracted increasing attention. Thus, according to recently published studies by Kisakov et al. [25,26], Abbasi et al. [27], and Wang et al. [28], the needle-free jet injection (JI) method is a promising alternative to LNPs for mRNA delivery. The method is based on the intradermal injection under high pressure, while the physical effects on the cell membrane provide increasing permeability to nucleic acids [29]. Additionally, the wide dispersion of the drug in the dermis facilitates the uptake of mRNA by antigen-presenting cells [30,31], which further increases its immunogenicity.

Hemagglutinin (HA) is the primary antigen for the development of influenza vaccines. Immunization with HA-based vaccines induces the production of neutralizing antibodies, which are crucial for defending against viral infection [32,33,34,35]. We have previously carried out the design of a modified HA trimer of influenza virus A/turkey/Stavropol/320-01/2020 (H5N8). The immunogen design is described in more detail in [36]. In brief, the transmembrane and cytoplasmic domains were excised from the native HA sequence. For the purpose of secretion from cells, the signal peptide from the native sequence was retained. In order to form stable homotrimers, the trimerizing domain of bacteriophage T4 fibritin was added to the C-terminus of the molecule. A number of amino acid substitutions were made to the native HA sequence with a view to stabilizing the protein. In our previous studies, we obtained recombinant HA/H5 protein and pVAX-H5 DNA vaccine encoding the designed immunogen. We showed that immunization of BALB/c mice with recombinant HA/H5 protein induced the formation of neutralizing antibodies with protective properties [37]. The pVAX-H5 DNA vaccine effectively induced specific humoral and cellular immune responses in laboratory animals [36,38] and protected BALB/c mice from lethal challenge with influenza virus strains A/chicken/Khabarovsk/24-1V/2022 (H5N1) and A/turkey/Stavropol/320-01/2020 (H5N8) [39]. It is important to note that we used the JI as a delivery system for the pVAX-H5 DNA vaccine.

The aim of this study was to develop an mRNA vaccine encoding a modified HA trimer of the influenza A/turkey/Stavropol/320-01/2020 (H5N8) virus and to analyze its immunogenic and protective properties when delivered by JI in comparison with LNPs.

## 2. Materials and Methods

### 2.1. Bacterial Strains, Viruses, Cell Cultures, and Plasmids

The *E. coli* strain Stbl3 (Invitrogen, Waltham, MA, USA) was used to produce plasmid DNA.

In the microneutralization assay, influenza virus strain A/turkey/Stavropol/320-01/2020 (H5N8) (EPI_ISL_1114749) (FBRI SRC VB «Vector» Rospotrebnadzor) and cell culture MDCK-SIAT1 (Cell culture collection of the FBRI SRC VB «Vector» Rospotrebnadzor, Koltsovo, Russia) were used.

The vaccine strain of influenza virus A/Astrakhan/3212/2020 (H5N8) [40] and strain A/chicken/Magadan/14-7V/2022 (H5N1) (EPI_ISL_16618968) (FBRI SRC VB «Vector» Rospotrebnadzor, Russia) [41] were used for the challenge study.

The antigenic properties of viruses (based on the hemagglutination inhibition (HI) test and microneutralization assay) are provided in the table (Table 1).

To obtain the DNA template, the recombinant plasmid pVAX-Cas1CC (FBRI SRC VB «Vector», Rospotrebnadzor) was used.

### 2.2. Preparation of DNA Template pVAX-Cas1CC-H5

The HA gene, designed based on the native HA gene of the influenza virus A/turkey/Stavropol/320-01/2020 (H5N8) (EPI_ISL_1114749), was cloned into the pVAX-Cas1CC expression cassette carrying the target gene under the control of the T7 promoter modified for efficient incorporation of the AG-Cap analog during synthesis. The cassette also contains the human α-globin 5′- and 3′-untranslated regions, as well as a 100-nucleotide poly (A) tail. Cloning was carried out using the recognition sites of restriction endonucleases Psp124BI and BamHI (SibEnzyme, Novosibirsk, Russia). The construct was named pVAX-Cas1CC-H5. The structure of the obtained plasmid was confirmed by restriction analysis and Sanger sequencing.

The DNA template was produced and purified using the “HiPure Plasmid Mini Kit” (Magen, Guangzhou, China) and linearized using the restriction endonuclease EcoRI (SibEnzyme, Novosibirsk, Russia).

### 2.3. In Vitro mRNA Synthesis

mRNA synthesis was performed using the linearized pVAX-Cas1CC-H5 template and a commercial kit (Yeasen, Shanghai, China). The reaction mixture included 1 μg of linearized DNA template, T7 polymerase with buffer, AG-Cap analog (m7GmAmG cap analog (Biolabmix, Novosibirsk, Russia) (6 mM), a mixture of ribonucleotide triphosphates (7 mM) with uridine replaced by N1-methylpseudouridine (Jena Bioscience, Jena, Germany), RNase inhibitor (BelBioLab, Moscow, Russia), and nuclease-free water. The synthesis protocol was described previously [26]. In brief, the protocol involved enzymatic synthesis and DNase treatment, followed by purification of the resulting product. The resulting mRNA was named mRNA-H5.

Analysis of the homogeneity and integrity of the obtained product, as well as its molecular size, was carried out by electrophoresis in a 2% agarose gel.

### 2.4. Lipid Nanoparticle Synthesis

LNPs were synthesized by rapid microfluidic mixing of an aqueous phase containing RNA and an ethanol (organic) phase containing a lipid mixture. The aqueous phase was a solution of mRNA-H5 in 100 mM citrate buffer, pH 4. The ethanol phase consisted of a mixture of ionizable lipid:phospholipid:helperlipid:PEG lipid in a molar percentage ratio of 50:10:38.5:1.5, respectively; all lipids were dissolved in 96% ethanol. The lipids included the following: ionizable lipid SM-102 (heptadecan-9-yl 8-((2-hydroxyethyl) (6-oxo-6-(undecyloxy)hexyl)amino)octanoate); phospholipid DSPC (1,2-distearoyl-sn-glycero-3-phosphocholine); auxiliary lipid cholesterol; and PEG lipid DMG-PEG2000 (1-monomethoxypolyethyleneglycol-2,3-dimyristylglycerol with polyethyleneglycol with an average molecular weight of 2000). The lipids were purchased from AVT Pharmaceutical Tech Co., China. The ratio of nitrogen in SM-102 to phosphate of mRNA was 6:1. The phases were mixed using the Automated NP System (Dolomite Microfluidics, Royston, UK). Mixing was carried out in a staggered herringbone micromixer microfluidic chip (Dolomite Microfluidics, Royston, UK). The procedure protocol is as follows: total mixture volume—1.6 mL; aqueous phase to organic phase ratio—3:1; and flow rate—3000 μL/min. A chip for diluting the LNP mixture with phosphate-buffered saline (PBS), pH 7, at a flow rate of 1000 µL/min was connected in series with the mixing chip.

Immediately after production, LNPs were diluted 20-fold with PBS, pH 7, and then concentrated using a JetSpin 30 k MWCO centrifugal concentrator (Jet Biofil, Guangzhou, China). The total volume of mRNA-H5-LNP obtained was then adjusted to obtain a final concentration corresponding to a dose of 10 or 30 μg of encapsulated mRNA-H5 in 100 μL of PBS. Before each immunization, a fresh series of mRNA-H5-LNP was prepared and stored at 4 °C for no more than 3 days.

For use as a control, LNPs without mRNA were obtained in a similar way, with citrate buffer as an aqueous phase.

### 2.5. Determination of Hydrodynamic Size and ζ-Potential of LNPs

To determine the charge and size of nanoparticles, dynamic and electrophoretic light scattering (DLS and ELS, respectively) methods were used, as described in [26]. In brief, the size of the nanoparticles was measured using the intensity-averaged particle size (Z-average) and polydispersity index (PDI), and the charge was measured using ζ-potential on a Zetasizer Nano ZS Plus (Malvern Instruments, Malvern, UK).

### 2.6. The Quantification of mRNA-H5-LNP Encapsulation

The encapsulation ratio and mRNA concentration were determined using the Quant-iTRiboGreen RNA reagent kit (Life Technologies, Waltham, MA, USA) according to the manufacturer’s protocol, except that mRNA-H5 was the standard RNA used to generate the calibration curve. Briefly, LNPs were degraded with 2% Triton X-100 in TE buffer in a 96-well plate. Next, 1× Quant-iTRiboGreen RNA dye in TE buffer was added to the sample, and the fluorescence data of the sample (λ_ex_ = 485 nm, λ_em_ = 520 nm) was measured using a Varioskan™ LUX plate reader (Thermo Fisher Scientific, Waltham, MA, USA). The total amount of nucleic acids in the sample was determined by applying the relative fluorescent units to the standard curve of nucleic acids with detergent. Simultaneously, to detect unencapsulated nucleic acid, intact LNPs were measured under the same conditions but without detergent. The amount of RNA encapsulated in LNPs was determined by subtracting the amount of unencapsulated RNA from the total amount of RNA in the sample. Finally, the amount of encapsulated RNA was divided by the total amount of RNA in the sample to obtain the encapsulation ratio.

### 2.7. Electron Microscopy of LNP Formulations

To determine the size and shape of mRNA-H5-LNP and LNPs without mRNA complexes, the suspension was applied to copper grids for electron microscopy coated with a carbon-stabilized film. The formulations were stained with 2% aqueous uranyl acetate. The analysis was carried out using a JEM-1400 electron microscope (Jeol, Tokyo, Japan). Image collection, analysis, and processing were performed using a Veleta digital camera (EMSIS GmbH, Münster, Germany).

### 2.8. Immunization of BALB/c Mice with mRNA-H5

The work with animals was carried out in accordance with the “Guide for the Care and Use of Laboratory Animals”. The protocols were approved by the Institutional Animal Care and Use Committee at the FBRI SRC VB «Vector» Rospotrebnadzor (Bioethics Committee Protocol No. 3 dated 29 February 2024). The mice were housed under a 12 h light/dark cycle with free access to food and water.

The study was conducted on inbred BALB/c mice, with an initial weight range of 16–18 g. The mRNA-H5-LNP formulations in 100 μL PBS were injected into the quadriceps muscle of the left hind paw using insulin syringes with a 29G needle. The mRNA-H5 preparations dissolved in 50 μL PBS were injected into the left hind paw using the JI method, as described previously [26].

The experiment was divided into 2 stages. In the first stage, to evaluate humoral and T-cell responses, the animals were divided into five groups of 12 animals each and immunized on days 0 and 21: Group 1 was immunized with 30 μg of mRNA-H5-LNP; Group 2 was immunized with 30 μg of mRNA-H5 using JI; Group 3 was immunized with 25 μg of recombinant HA/H5 protein with aluminum hydroxide; Group 4 was immunized with LNPs without mRNA (the amount of LNPs was equivalent to Group 1); and Group 5 consisted of non-immunized animals. A dose of 30 μg mRNA was selected based on the findings of prior studies [25,26] and the results of JI studies documented in the literature [27]. On day 35, blood was collected from the retroorbital sinus of the eye. Animals were humanely euthanized by cervical dislocation, followed by spleen collection for T-cell response analysis.

At the second stage, the animals were immunized to analyze the protective effect of the mRNA vaccine. Consistent with the observed adverse effects of the high dose of LNPs in the first stage of the experiment, the dose of mRNA-H5-LNP in the second stage was reduced to 10 μg. The animals were divided into 8 groups of 10 animals each and immunized on days 0 and 21: Groups 1 and 2 were immunized with 10 μg of mRNA-H5-LNP; Groups 3 and 4 were immunized with 30 μg of mRNA-H5 using JI; Groups 5 and 6 were immunized with 25 μg of recombinant HA/H5 protein with aluminum hydroxide; and Groups 7 and 8 consisted of non-immunized animals. The number of animals in each group was determined based on the findings of prior studies of a similar immunogen in the form of a DNA vaccine [36,39]. On day 35, blood was collected from the retroorbital sinus of the eye to analyze the humoral immune response.

### 2.9. Enzyme-Linked Immunosorbent Assay (ELISA)

ELISA was performed according to the method described in [36,42]. Recombinant HA/H5 protein of the influenza virus A/turkey/Stavropol/320-01/2020 (H5N8) was used as an antigen [37]. In brief, sera of immune animals were used as primary antibodies (diluted 1:10). Goat antibodies against mouse IgG labeled with horseradish peroxidase (Sigma-Aldrich, St. Louis, MO, USA) were used as secondary antibodies. Tetramethylbenzidine solution (IMTEK, Moscow, Russia) was used as a chromogenic substrate. The reaction was stopped with 1 N hydrochloric acid solution. Optical density was measured on a Varioskan LUX device (Thermo Fisher Scientific, Waltham, MA, USA) at a wavelength of 450 nm. The serum dilution at which the optical density value was more than twice that of the negative control (in which blocking buffer was added to the wells instead of serum) was the last dilution that gave a positive response. The final titer was determined by the last diluted sample that gave positive results for ELISA.

### 2.10. In Vitro Microneutralization Assay

The microneutralization assay was carried out according to the method described in [37]. In brief, animal sera were treated with RDE (Denka Seiken, Tokyo, Japan), a receptor-destroying enzyme, according to the protocol recommended by the WHO [43]. The standard virus dose was 100 TCID_50_/100 μL of viral diluent. Two-fold dilutions of blood serum were prepared in 200 μL of viral diluent in a 96-well plate, then the same volume of standardized viruses was added to each well, and the suspension was incubated for 1 h at 37 °C and 5% CO_2_. The naive mouse serum was used as a negative control. The reference ferret serum anti-A/turkey/Stavropol/320-01/2020 (H5N8) (FBRI SRC VB «Vector» Rospotrebnadzor, Novosibirsk, Russia) was used as a positive control. After that, 200 μL of the suspension was transferred to the wells of a 96-well plate with a 90% monolayer of MDCK-SIAT1 cell culture; the plates were incubated for 70–72 h in Opti-MEMI medium with 1 μg/mL TPCK-trypsin (Sigma-Aldrich, St. Louis, MO, USA) and antibiotic (Anti-Anti, Gibco, Waltham, MA, USA) at 37 °C, 5% CO_2_. After incubation, the cells were stained with crystal violet solution and analyzed using an Agilent BioTek Cytation 5 multi-mode cell imaging device (Thermo Fisher Scientific, Waltham, MA, USA). If no more than 5% of living cells remain in the negative control, the titer of the test serum is equal to the last dilution at which 50% of living cells remain.

### 2.11. Assessment of T-Cell Response by Intracellular Cytokine Staining (ICS)

On day 35, after the first immunization, spleens were harvested from mice. Splenocytes were obtained by pushing the spleen through 70 and 40 μm filters. After erythrocyte lysis, remaining cells were washed twice in RPMI medium and then resuspended in RPMI containing 2 mM L-glutamine and 50 μg/mL gentamicin.

Splenocytes were seeded in 96-well round-bottomed plates (Jet Biofil, China) at 7 × 10^5^ cells per well; a peptide pool (20 μg/mL for each peptide), medium (as a negative control), or a mixture of mitogens (phorbol 12-myristate 13-acetate (PMA) (30 ng/mL) and ionomycin (1 μg/mL)) (as a positive control) was added.

Peptides from the HA of the influenza A/H5N8 virus used to stimulate splenocytes include the following: MPFHNIHPL, AGWLLGNPM, CYPGSLNDY, RVPEWSYIV, LRNSPLREKRRKRGL, YVKSNKLVL, TYNAELLVL, LYDKVRLQL, SFFRNVVWL, SPYQGAPSF, LYKNPTTYISVGTSTLNQ, VDTIMEKNVTVTHAQDILEK, and SSWPNHETSLGVSAASPYQ. The peptides have a high probability of binding to MHC H-2D of BALB/c mice (determined using the NetMHCpan-4.1 service).

Cells were incubated for 3 h at 37 °C in 5% CO_2_ and for an additional 15 h with brefeldin A (5 μg/mL, GolgiPlug, BD Biosciences, Franklin Lakes, NJ, USA). The next day, cells were washed and stained with anti-CD3 Alexa Fluor 700 (clone 500A2, Biolegend, San Diego, CA, USA), anti-CD4 BV785 (clone GK1.5, Biolegend, San Diego, CA, USA), and anti-CD8 FITC (clone 53-6.7, Biolegend, San Diego, CA, USA). Cells were fixed with 1% paraformaldehyde for 10 min, washed, and permeabilized with 0.2% Tween-20 for 10 min. Afterwards, cytokine staining was performed with anti-IFNγ APC (clone XMG1.2, Biolegend, San Diego, CA, USA), anti-IL-2 BV 421 (clone JES6-5H4, Biolegend, San Diego, CA, USA), and anti-TNFα PE (clone MP6-XT22, Biolegend, San Diego, CA, USA). After 20 min, the cells were washed, dissolved in 1% paraformaldehyde solution, and incubated for 20 min, followed by analysis on a ZE5 flow cytometer (Bio-Rad, Hercules, CA, USA) using the Everest program.

### 2.12. Virulence

To determine the virulence of avian influenza viruses, female BALB/c mice weighing 18–20 g were infected intranasally. The mice were divided into six groups of six animals each to determine the 50% lethal dose (MLD_50_). The animals were lightly anesthetized with a combination of Zoletil 100 (Delpharm Tours, Tours, France) and Xyla (Interchemie, Venray, The Netherlands) before intranasal inoculation with 0.05 mL of the virus diluted 10-fold. The six groups were observed for 14 days to detect clinical signs and mortality. The 50% lethal dose in mice was calculated using the Reed–Munch method.

### 2.13. Challenge Study

All challenge studies were carried out in compliance with the requirements of SanPiN 3.3686-21, “Sanitary and epidemiological requirements for the prevention of infectious diseases” [44].

Mice were infected intranasally with 20 MLD_50_ (MLD_50_ = 4.4 ± 0.4 lgEID_50_ (embryonic infectious dose)) of the influenza virus strain A/Astrakhan/3212/2020 (H5N8) or 20 MLD_50_ (MLD_50_ = 2.0 ± 0.4 lgEID_50_) of the influenza virus strain A/chicken/Magadan/14-7V/2022 (H5N1). Mice were inoculated under anesthesia with a mixture of tiletamine and zolazepam, as well as xylazine chloride. The animals were observed daily after infection for 14 days; clinical signs indicating the development of the disease (ruffled fur, hypothermia, exhaustion, neurological disorders, and death) were monitored, and the weight of the animals was measured. In cases where mice developed severe conditions incompatible with life, such as anorexia (loss of more than 20% of the initial body weight) or lethargy, the animal was humanely euthanized by cervical dislocation. All other animals were humanely euthanized by cervical dislocation at the end of the experiment.

### 2.14. Statistical Analysis

Statistical data processing was performed using GraphPad Prism 9.0 software (GraphPad Software, San Diego, CA, USA). Quantitative data are provided as the mean with a range of values or a range from lowest to highest and analyzed using nonparametric tests. Intergroup differences were assessed using nonparametric one-way Kruskal–Wallis analysis of variance with correction for multiple comparisons and Dunn’s statistical hypothesis testing. Survival function modeling was performed using the Kaplan–Meier multiplicative estimator, and survival comparison with the control group was performed using the Mantel–Cox test. Comparisons were not statistically significant unless otherwise stated.

## 3. Results

### 3.1. Preparation of the DNA Template and mRNA Synthesis

The HA gene, encoding the HA of the HPAI virus subtype A/H5N8, which was circulating in Russia in Stavropol in 2020 (A/turkey/Stavropol/320-01/2020), described in [36], was designed and cloned into the pVAX-Cas1CC expression cassette, resulting in the DNA template pVAX-Cas1CC-H5. Synthesis of the target mRNA (Figure 1a) was then carried out from the linearized DNA template; the synthesis products were verified by electrophoresis (Figure 1b). The size of the resulting mRNA was shown to be the same as the theoretical calculation (1900 nucleotides), and the purity of the synthesis product was demonstrated. The whole electrophoregram is shown in Appendix A.

### 3.2. Lipid Nanoparticle Synthesis and Characterization

The mRNA-H5 obtained was then encapsulated in LNPs for the subsequent immunization of mice. For use as a control, LNPs without mRNA were obtained in a similar way.

The obtained formulations were characterized using DLS and ELS methods. The DLS method (Table 2) was used to analyze the samples, revealing that both the mRNA-H5-LNP complex and the LNPs without mRNA resulted in a monodisperse suspension of nanoparticles (PdI = 0.149 ± 0.01 and 0.175 ± 0.01, respectively). The mean hydrodynamic diameter of mRNA-H5-LNP was 93.5 ± 0.8 nm, while for LNPs without mRNA, this parameter was higher at 131.6 ± 1.6 nm. This discrepancy may be explained by the spontaneous random self-assembly of nanoparticles in the absence of mRNA, which can result in an increase in mean size [45]. Characterization data for formulations obtained for subsequent immunizations are not provided, since the deviation from the presented data for each of the subsequent formulations was less than 10% according to DLS and ELS data (Table A1).

The particle distribution profile (Figure 2b) demonstrates the presence of a single peak, indicating the absence of fractions with other particle sizes and the absence of impurities.

The ζ-potential value of the mRNA-H5-LNP particles in three series of measurements was −0.02 ± 0.26 mV, in line with theoretical calculations. The almost zero charge indicates that the negatively charged mRNA molecules were almost completely packed into the LNPs. This was confirmed by Quant-iTRiboGreen RNA analysis, which also showed that an encapsulation efficiency of more than 90% was achieved. Electron microscopy (Figure 2a) confirmed that nanoparticles of the required size had formed.

### 3.3. A Comparison of the Immunogenicity of mRNA-H5 Administered to BALB/c Mice Using LNPs and JI

BALB/c mice were immunized to analyze the immunogenicity of mRNA-H5 and compare the efficacy of the aforementioned delivery methods.

ELISA analysis (Figure 3) showed that specific antibody titers were detected in the experimental groups of animals that were immunized with mRNA-H5 delivered by LNPs (average titer was 5.74 lg10—sera dilution 1:546,800) and by JI (average titer was 5.21 lg10—sera dilution 1:164,000). Specific antibody titers were also detected in the group that was immunized with recombinant HA/H5 protein with aluminum hydroxide (average titer was 5.26 lg10—sera dilution 1:182,300).

Microneutralization analysis showed that the sera of animals immunized with mRNA-H5 delivered by LNPs, mRNA-H5 delivered by JI, and recombinant HA/H5 protein are capable of neutralizing live influenza virus strain A/turkey/Stavropol/320-01/2020 (H5N8); the average 50% neutralization titer was 3.76 lg10, 3.23 lg10, and 3.48 lg10 (sera dilution 1:5700, 1:1700, and 1:3000), respectively.

The ICS analysis demonstrated that immunization with mRNA-H5 resulted in the generation of both CD3^+^/CD8^+^ and CD3^+^/CD4^+^ lymphocytes, which were capable of producing IFNγ, IL-2, and TNFα after stimulation with HA-specific peptides (Figure 4).

### 3.4. Study of the Ability of mRNA-H5 Delivered by LNPs and JI to Protect Mice from Lethal Challenge with Avian Influenza A/H5 Viruses

After the B- and T-cell response induced by the mRNA-H5 had been analyzed, the next step was to evaluate its protective potential. The dose of mRNA-H5-LNP was reduced in accordance with the principles of humanity.

Mice were immunized twice at doses of 10 and 30 µg for LNPs and JI, respectively. The animals immunized with the recombinant HA/H5 protein and the naive animals were used as controls. The resulting titers (Figure 5a) estimated by ELISA were 5.82 lg10 (sera dilution 1:656,100) for mRNA-H5-LNP and 5.24 lg10 (sera dilution 1:175,000) for mRNA-H5 JI. The recombinant HA/H5 protein was found to stimulate a humoral immune response at a titer of 5.29 lg10 (sera dilution 1:196,800).

Then, on day 35, after the first immunization, the mice in the 1, 3, 5, and 7 groups were infected intranasally with the influenza virus strain A/Astrakhan/3212/2020 (H5N8) [40], and the mice in the 2, 4, 6, and 8 groups were infected intranasally with A/chicken/Magadan/14-7V/2022 (H5N1) [41]. The results (Figure 5b,c) showed that the survival rate in groups of mice immunized with mRNA-H5-LNP and mRNA-H5 JI when infected with both strains was 100%. There were no signs of disease or weight loss in the animals in the experimental groups. The survival rate in the negative control group (naive animals) was 0%; as early as the fifth day after infection, neurological symptoms (e.g., decreased activity, loss of appetite) and ruffling were observed in animals, as well as significant weight loss (Figure 5d,e). The survival rate in the group of animals immunized with the recombinant HA/H5 protein was 100% in the case of infection with a homologous strain and 90% in the case of infection with a heterologous strain.

## 4. Discussion

The use of the mRNA platform for the development of vaccines against infection with HPAI A/H5 viruses has been described in numerous articles [16,46,47,48]. In these studies, LNPs are used as a method of mRNA vaccine delivery. LNPs have demonstrated efficacy in delivering mRNA into the cytosol of target cells while protecting it from degradation within the body [49]. However, it should be noted that certain components of LNPs have the potential to cause adverse effects. For instance, the presence of helper lipids has been observed to induce systemic inflammation through their absorption by Kupffer cells within the liver [50]. Furthermore, the interaction of certain helper lipids with phospholipases A_2_ has been demonstrated to result in the production of allergens, which can subsequently trigger complement activation and anaphylaxis [51,52]. It must be noted that the aforementioned list does not include all the problems reported with the use of LNPs [53,54,55,56,57]. In our studies, we employ a composition analogous to that of Moderna and Pfizer, which was developed for the purpose of vaccine development against the SARS-CoV-2 virus. The aforementioned disadvantages pertain exclusively to the lipids used in this formulation. The present focus of research endeavors [58,59,60,61] is on the identification of an alternative formula that would curtail reactivity while preserving adequate immunogenicity.

An alternate approach to mRNA vaccine delivery involves JI, a method that lacks the disadvantages associated with LNPs [25,27]. In a previous study [26], it was demonstrated that the delivery of naked mRNA-RBD using a spring-loaded jet injector resulted in higher vaccine immunogenicity compared to the conventional intramuscular injection method that utilizes a needle and syringe. It was observed that naked mRNA-RBD delivered by JI does not spread systemically, resulting in reduced vaccine reactogenicity and adverse effects.

In this study, we have developed an experimental mRNA vaccine, mRNA-H5, encoding a modified HA trimer of influenza virus A/turkey/Stavropol/320-01/2020 (H5N8). Analysis of the humoral immune response, conducted at the first stage of the study, indicated that the group immunized with mRNA-H5-LNP exhibited a statistically significant increase in the level of humoral immune response in comparison with the group immunized with mRNA-H5 using JI. These data suggest that the encapsulation of mRNA in LNP may provide more efficient mRNA delivery to immune system cells. A factor that could influence the immunogenicity of mRNA delivered by JI is the partial degradation of the naked mRNA molecule by extracellular nucleases [49].

Cellular immunity also plays an important role in the prevention of influenza. The results obtained demonstrated that the percentage of CD4+ and CD8+ T cells that secrete IFNγ, IL-2, and TNFα was significantly higher in mice immunized with mRNA-H5 than in negative control groups (Figure 4). T-cells are incapable of providing sterilizing immunity against influenza; nevertheless, they frequently offer broader protection against various strains or subtypes of the virus [62]. This property is a significant component in the induction of comprehensive immune responses. It is also important to note that there were no statistically significant differences in IFNγ, IL-2, and TNFα production between the groups immunized with both JI and LNPs.

While the immune response in mice immunized with mRNA-H5-LNP (30 µg) was higher than in those immunized with mRNA-H5 JI, the condition of the animals in the first group was unsatisfactory. It was observed that the animals immunized with mRNA-H5-LNP exhibited signs of distress, including coat ruffling, as well as heightened discomfort during the second immunization procedure; these phenomena were not observed in the mRNA-H5 JI group. These effects may be a consequence of an inflammatory process induced by a substantial number of LNPs [63,64]. A similar phenomenon was observed in studies by Mulligan et al. [65] when investigating various doses of the BNT162b1 vaccine in humans. When higher doses were used, adverse effects such as fever, pain at the injection site, and malaise were observed. These adverse effects have not yet been studied in detail but will be examined in future studies.

In consideration of the above, the protocol for evaluating the vaccine’s protective efficacy was modified to reduce a dosage of mRNA-H5-LNP, set at 10 µg, in accordance with the principles of humanity. The antibody titer induced by mRNA-H5-LNP was similar within the range of error at doses of 10 (5.82 lg10) and 30 μg (5.74 lg10), according to the analysis of the humoral immune response. A slight difference in antibody titers may be explained by a number of factors. According to the studies of several researchers, the cells have limited capacity to translate exogenous mRNA into protein. It has been determined that it is possible to achieve the maximum antigen production at a dose of 10 µg of mRNA-LNPs complex and that an increase of a dose does not result in a higher level of expression after the second immunization [65,66,67]. Another factor is the buffering capacity of nanoparticles. They have limited ability to effectively deliver drugs into cells due to saturable endocytosis and limited intracellular release associated with nanoparticle fusion and depletion of the available receptors [68,69,70,71]. Furthermore, research conducted by Qin et al. [72] demonstrates that the repeated administration of mRNA-LNPs to the same area as the initial administration results in a reduction in the immune response, which may be dose-dependent. The detection of this phenomenon indicates the necessity for further exploration of the dose-dependent effect of the mRNA-H5.

A challenge study demonstrated that immunization with both naked mRNA-H5 administered with JI and mRNA-H5 encapsulated in LNPs provided 100% protection of animals against lethal challenge of both strains of avian influenza virus (A/Astrakhan/3212/2020 (H5N8) and influenza A/chicken/Magadan/14-7V/2022 (H5N1)) (Figure 5). The vaccine’s ability to provide protection against both homologous and heterologous HPAI viruses may indicate its broad-spectrum protection. One potential explanation for the complete protection of immunized mRNA-H5 mice against heterologous challenge is the presence of sufficient amino acid sequence similarity between the HA in heterologous strains of avian influenza virus A/turkey/Stavropol/320-01/2020 (H5N8) and A/chicken/Magadan/14-7V/2022 (H5N1). The degree of homology between the HA of the specified A/H5N8 and A/H5N1 strains was determined, revealing a high degree of similarity, with one amino acid substitution (V11I) in A/chicken/Magadan/14-7V/2022 (H5N1) in the head domain of HA. This domain is responsible for binding to host cell receptors and is a target for virus-neutralizing antibodies.

As illustrated in Table 1, the antigenic properties of the A/chicken/Magadan/14-7V/2022 (H5N1) and A/Astrakhan/3212/2020 (H5N8) viruses exhibit minimal variation. The titers of reference serum with these viruses in the HI test and in microneutralization assay differ by no more than fourfold. Therefore, the cross-protection against H5N1 is attributed to HA homology.

Additionally, our earlier study of a DNA vaccine encoding an immunogen similar to that in mRNA-H5 [39] yielded data on cross-reactivity against heterologous strains. The pVAX-H5 DNA vaccine was shown to provide 100% protection against the HPAI virus strain A/chicken/Khabarovsk/24-1V/2022 (H5N1) in mice. This strain has significant antigenic differences from the one on which the immunogen was based. This fact may indicate the immunogen’s ability to provide broad protection against various virus strains.

The employment of mRNA delivery strategies, in conjunction with the administration of recombinant protein with aluminum hydroxide, resulted in a high level of protection of mice against infection with two different strains of influenza virus. However, the JI appears to be the most promising in this context, as it is simpler and more economical. A comparison of mRNA and protein as immunogens reveals that mRNA is a more promising approach, as it does not necessitate complex purification procedures and is synthesized in a cell-free system.

Furthermore, a study by Abbasi et al. [27] demonstrated that JI of naked mRNA elicited a robust immune response in mice and non-human primates, with spike-specific IgG levels persisting at elevated levels for over six months following booster therapy. This finding indicates the presence of persistent humoral immune responses after JI of naked mRNA. The authors observe that the two approaches, LNPs and JI, result in comparable humoral immune responses with minimal systemic inflammation, suggesting the progressive effectiveness of these methods in immune response stimulation, despite differences in delivery mechanisms. In addition, Wang et al. [28] evaluated the safety and immune response to the RBD3-Fc vaccine administered as naked mRNA using JI compared to mRNA-LNP. The prime-boost immunization protocol elicited a robust immune response, and intradermal administration demonstrated comparable or even superior outcomes in antibody binding, neutralizing antibodies, and T-cell response. The efficacy of naked mRNA vaccines was found to be inferior to that of mRNA-LNP vaccines. Moreover, the presence of long-term antibody persistence was detected four weeks after booster therapy, suggesting the formation of memory B cells. The study demonstrated that rats vaccinated with mRNA-LNP using JI elicited robust systemic immune responses, elevated virus-neutralizing antibody titers, and substantial Th1 and Th2 immune responses. However, a direct comparison of the duration of the immune response between mRNA-LNP and JI of naked mRNA under identical conditions was not performed. Consequently, it can be concluded that both methods of mRNA vaccine delivery demonstrate efficacy at therapeutic doses, thereby opening up opportunities for further research into the relative durability of immune responses.

Overall, the obtained mRNA vaccine is a promising candidate for use as a vaccine against infection with HPAI A/H5 viruses. A relevant area of further research is to evaluate the dose-dependent effect of mRNA delivered by both LNPs and JI and estimate the protective potential of the vaccine against infection with other heterologous A/H5 viruses.

However, it is important to acknowledge that although mice are the most commonly used animal model for the preliminary assessment of influenza vaccines, they do not naturally carry the virus, in contrast to humans. The predictive value of results obtained in mice may be affected by differences in disease manifestations, immune response, and virus transmission characteristics when evaluating vaccine efficacy in humans.

Within a year, we plan to conduct mRNA-H5 trials on ferrets, which are a more relevant model. Prior to this, a comprehensive analysis of the vaccine is planned, which will be conducted on smaller animal models (e.g., BALB/c mice). This analysis will include a more precise determination of the optimal dose for each vaccine variant. Immunization of ferrets with mRNA vaccine in LNP and with JI and with recombinant HA/H5 protein is planned. The immune response will be evaluated using ELISA, microneutralization assay, and ELISpot. The ability of mRNA-H5 to protect animals (ferrets) from infection with HPAI viruses A/H5N8 and A/H5N1 strains from disparate geographic regions will also be investigated. In addition, the following procedures will be carried out: quantification of viral RNA in ferret organ samples, pathological analyses of lung tissue, and assessment of influenza virus reproduction in the lungs. The selection of the model is predicated on the potential for observing clinical manifestations or the absence thereof, similar to those observed in humans, that correspond to the influenza. Furthermore, we plan to study the adverse effects discussed in our current study based only on the literature sources. The study will include an assessment of toxicity, allergenic properties, mutagenicity, histopathological changes, and a number of other indicators using guinea pigs and/or rabbits as model animals.

## 5. Conclusions

Consequently, in the current study, an experimental mRNA vaccine encoding a modified HA trimer of the influenza A/turkey/Stavropol/320-01/2020 (H5N8) virus was developed. Immunization of BALB/c mice with the obtained mRNA vaccine using LNPs and JI has been shown to effectively stimulate specific humoral and T-cell immune responses. An analysis of delivery methods demonstrated that mRNA-H5 encapsulated in LNPs resulted in a more significant stimulation of the humoral response compared to mRNA-H5 delivered by JI at the same dose. However, the level of humoral response to mRNA-H5 JI was comparable to the response to recombinant HA/H5 protein with aluminum hydroxide at a dose of 25 μg. Moreover, mRNA vaccine mRNA-H5 delivered by LNPs and JI provided 100% protectivity of immunized mice against lethal challenge with homologous and heterologous strains of avian influenza virus (A/Astrakhan/3212/2020 (H5N8) and A/chicken/Magadan/14-7V/2022 (H5N1), respectively). The present results indicate that JI can be considered as an alternative to LNPs for mRNA delivery. It is necessary to study the dose-dependent effects of mRNA-H5-LNP and mRNA-H5 JI, as well as confirm the protective properties of the vaccine in other relevant animal models, such as ferrets.

mRNA-H5 may be considered as a promising candidate for the development of a vaccine against infection with HPAI A/H5 viruses with pandemic potential.

## Figures and Tables

**Figure 1 vaccines-13-00883-f001:**
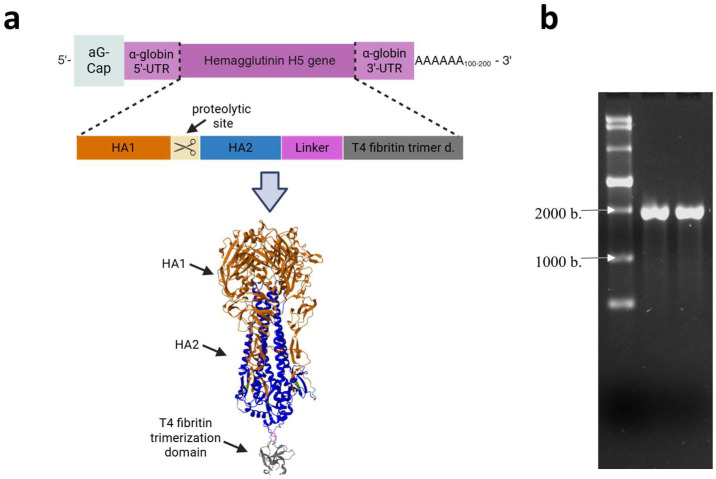
Structure and production of the mRNA-H5 vaccine. (**a**) Schematic representation of mRNA: mRNA has a 5′-cap, poly (A) tail, 5′- and 3′-untranslated regions of human α-globin, and an open reading frame encoding the hemagglutinin (HA) of the highly pathogenic avian influenza (HPAI) A/H5N8 virus; schematic structure of the modified HA gene; the model of the HA trimer encoded by mRNA-H5 (the model was obtained as described in [37]). (**b**) Electrophoregram of mRNA-H5 synthesis products in 1% agarose gel: lane 1—ssRNA Ladder (New England Biolabs); lanes 2 and 3—mRNA-H5.

**Figure 2 vaccines-13-00883-f002:**
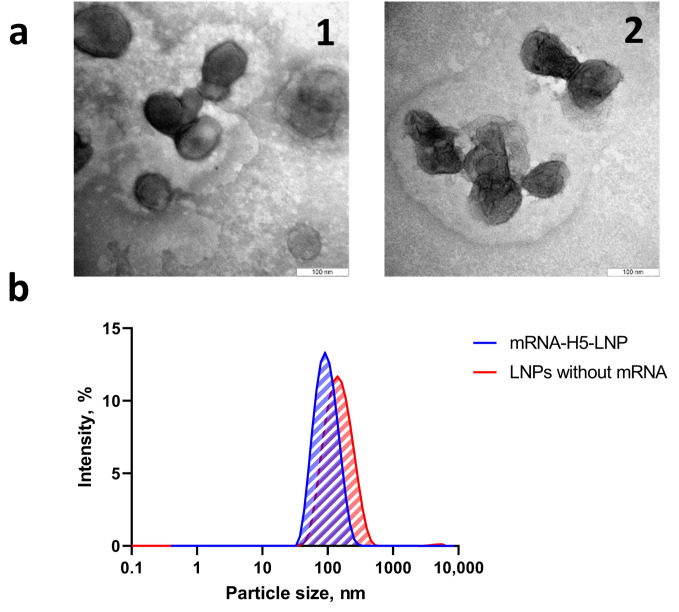
Preparation and characterization of LNP formulations. (**a**) Electron micrograph of complexes: (**1**) mRNA-H5-LNP; (**2**) LNPs without mRNA. (**b**) Dynamic light scattering: size distribution profiles of nanoparticles obtained with and without mRNA.

**Figure 3 vaccines-13-00883-f003:**
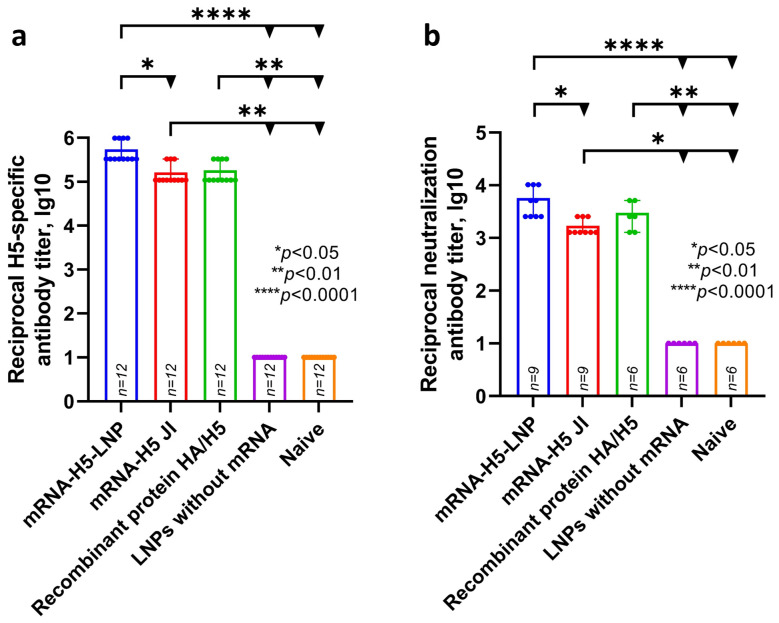
Study of humoral response to mRNA-H5. (**a**) H5-specific antibody titer (ELISA data) (number of animals: *n* = 12). (**b**) Microneutralization antibody titers against influenza virus A/turkey/Stavropol/320-01/2020 (H5N8) (number of animals in mRNA-H5 groups: *n* = 9; in other groups: *n* = 6). Reciprocal titer values are provided in the plots. mRNA-H5-LNP—a group of animals immunized with mRNA-H5 delivered by LNPs; mRNA-H5 JI—a group of animals immunized with mRNA-H5 delivered by needle-free jet injection (JI); Recombinant protein HA/H5—a group of animals immunized with recombinant protein HA/H5; LNPs without mRNA—a group of animals immunized with LNPs without mRNA; Naïve—a group of non-immunized animals. In panels, data are provided as means with full range. * *p* < 0.05, ** *p* < 0.01, **** *p* < 0.0001 following Kruskal–Wallis analysis of variance with correction for multiple comparisons and Dunn’s statistical hypothesis test.

**Figure 4 vaccines-13-00883-f004:**
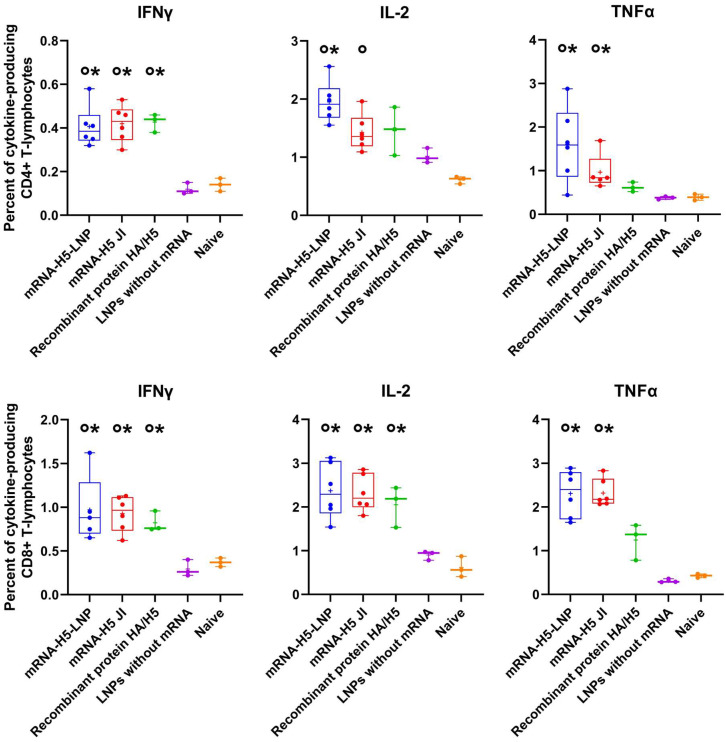
Percentage of HA-specific cytokine-producing CD3^+^/CD4^+^ and CD3^+^/CD8^+^ lymphocytes analyzed using the intracellular cytokine staining (ICS) method combined with flow cytometry (number of animals: *n* = 6). mRNA-H5-LNP—a group of animals immunized with mRNA-H5 delivered by LNPs; mRNA-H5 JI—a group of animals immunized with mRNA-H5 delivered by JI; Recombinant protein HA/H5—a group of animals immunized with recombinant protein HA/H5; LNPs without mRNA—a group of animals immunized with LNPs without mRNA; Naive—a group of non-immunized animals. The data is provided in the form of a “box-and-whiskers plot”. The box shows the interquartile range, and the whiskers show the full range of values. Individual values are represented by dots; group medians—by lines; mean values—by pluses. *p*< 0.05 compared ^0^ with LNPs without mRNA; * with naive mice.

**Figure 5 vaccines-13-00883-f005:**
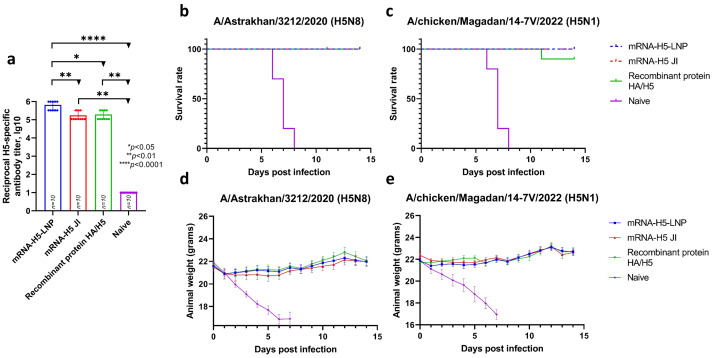
Analysis of mRNA-H5 protective properties (number of animals: *n* = 10). (**a**) Analysis of humoral immune response before challenge. H5-specific antibodies (ELISA data). Data are provided as mean with range. * *p*< 0.05; ** *p*< 0.01; **** *p*< 0.0001 following Kruskal–Wallis analysis of variance with correction for multiple comparisons and Dunn’s statistical hypothesis test. Data are provided as mean with full range. (**b**,**c**) Survival curves of immunized animals after infection with the influenza virus strain: (**b**) A/Astrakhan/3212/2020 (H5N8); (**c**) A/chicken/Magadan/14-7V/2022 (H5N1). (**d**,**e**) Weight change curves of mice in a challenge study against strain: (**d**) A/Astrakhan/3212/2020 (H5N8); (**e**) A/chicken/Magadan/14-7V/2022 (H5N1). Data are provided as mean with standard error of the mean. mRNA-H5-LNP—a group of animals immunized with mRNA-H5 delivered by LNPs; mRNA-H5 JI—a group of animals immunized with mRNA-H5 delivered by JI; Recombinant protein HA/H5—a group of animals immunized with recombinant protein HA/H5; Naive—a group of non-immunized.

**Table 1 vaccines-13-00883-t001:** Antigenic properties of viruses used in the study.

Viruses	Titer with Anti-A/Astrakhan/3212/2020 Ferret Reference Serum	
	HI-Test	Microneutralization Assay
A/Astrakhan/3212/2020 H5N8	640	320
A/turkey/Stavropol/320-01/2020 H5N8 2.3.4.4b	320	80
A/chicken/Magadan/14-7V/2022 H5N1 2.3.4.4b	160	80

**Table 2 vaccines-13-00883-t002:** Physical characteristics of the obtained lipid nanoparticles (LNPs) formulations.

Formulations	Average Particle Size, nm	Polydispersity Index	ζ-Potential
mRNA-H5-LNP	93.5 ± 0.8	0.149 ± 0.01	−0.02 ± 0.26
LNPs without mRNA	131.6 ± 1.6	0.175 ± 0.01	1.15 ± 0.26

## Data Availability

The data can be shared upon request.

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
