# Peer review of "Immunogenic and Protective Properties of mRNA Vaccine Encoding Hemagglutinin of Avian Influenza A/H5N8 Virus, Delivered by Lipid Nanoparticles and Needle-Free Jet Injection"

_vaccines, 2025, doi:10.3390/vaccines13080883_

Round 1

Reviewer 1 Report

Comments and Suggestions for Authors

In this manuscript the authors tested an experimental mRNA vaccine mRNA-H5, which encodes a modified hemagglutinin trimer of influenza virus A/turkey/Stavro-25 pol/320-01/2020 (H5N8). Authors explain the methodology for this development which uses lipid nanoparticles (LNPs) and tested it (after several controls) as an immunogen; Another group of animals only received mRNA (transcribed with special reagents) by a needle-free jet injection approach (subcutaneous). Antibody and cellular responses were evaluated with both using as controls a group of animals that only received the LNPs without mRNA and another group immunized with the purified HA protein (modified as described in a previous paper). A challenge experiment was also done and the results showed that both strategies provided 100% protection. However, since the LNP approach may induce secondary problems, the authors suggest that the needle-free jet injection may be a more suitable alternative.

I find the manuscript very interesting. The authors have published several papers with similar structure, testing different influenza immunogenic approaches (plasmid, proteins, etc). This time they developed a mRNA vaccine delivered combined with LNPs or naked. The immune and challenge responses suggest a similar performance of the

These are my questions/ suggestions to improve visibility of the results and to clarify potential doubts of the readers

-Although the authors made a reasonable description of the methods, it would be enriching to add, when possible, a brief comment of the procedure instead of mentioning “as before” or as “published”. That includes a brief description of the modified HA/H5 and the pVAX-H5 DNA plasmid.

-There should be a comment about the criteria for the chosen doses used for each procedure.

-Why the purified HA/H5 protein was used for comparison and not the plasmid pVAX-H5 DNA.

-Discussion repeats too much of the results. Perhaps it would be better to add information/comparisons with similar strategies: all types of lipid vesicles used for mRNA delivery are like the ones used in this manuscript? all of them may  generate complications? If protection is 100% for all strategies used as shown in the graphics (included recombinant protein), which would be the best in terms of cost and simplicity of elaboration?

Author Response

We would like to thank the reviewer for careful and thorough reading of our manuscript and for the thoughtful comments and constructive suggestions, which help to improve its quality. Our response follows.

Corrections to the text of MS are highlighted in green.

[Comments 1]: Although the authors made a reasonable description of the methods, it would be enriching to add, when possible, a brief comment of the procedure instead of mentioning “as before” or as “published”. That includes a brief description of the modified HA/H5 and the pVAX-H5 DNA plasmid.

[Response 1]: Agree. We have, accordingly, modified, when possible, "Introduction" and "Materials and methods", to emphasize this point.

[Comments 2]: There should be a comment about the criteria for the chosen doses used for each procedure

[Response 2]: We added the information to “Materials and Methods”:

«A dose of 30 μg mRNA was selected based on the findings of prior studies [29,30] and the results of jet injection studies documented in the literature [31]. »

and

«Consistent with the observed adverse effects of the high dose of LNPs in the first stage of the experiment, the dose of mRNA-H5-LNP in the second stage was reduced to 10 μg».

[Comments 3]: Why the purified HA/H5 protein was used for comparison and not the plasmid pVAX-H5 DNA?

[Response 3]: In most cases, vaccines against seasonal influenza, as well as vaccines against HPAI virus infection, are protein-based. It is noteworthy that both inactivated and subunit vaccines contain substantial quantities of HA. Therefore, it was decided that purified HA/H5 protein would be used as a positive control.

It was important to demonstrate that mRNA encoding the HA/H5 gene produces an adequate amount of HA protein in the body to elicit an immune response similar to that induced by the HA protein's intramuscular administration.

[Comments 4]: Discussion repeats too much of the results. Perhaps it would be better to add information/comparisons with similar strategies: all types of lipid vesicles used for mRNA delivery are like the ones used in this manuscript? all of them may generate complications? If protection is 100% for all strategies used as shown in the graphics (included recombinant protein), which would be the best in terms of cost and simplicity of elaboration?

[Response 4]: Agree. We have, accordingly, revised "Discussion".

Reviewer 2 Report

Comments and Suggestions for Authors

I read  this MS, which is clearly presented, with interest. The authors describe murine studies to evaluate an mRNA vaccine for avian influenza, in which the describe vaccine formulations, immunisation studies and finally an efficacy study.  This Ms will be of interest to other researchers using mRNA vaccine approaches and addresses the current concern over the spread of avian influenza. I have some suggestion to improve the MS:

Line 29: Protectivity is more usually 'protection' or 'efficacy'

Lines 191-200 Methods: can the authors say why they chose different dose-levels of mRNA for the 3 key immunisation groups? did they do a previous dose ranging study?

Ines 262: Did the authors calculate he MLD50 for both the homologous and heterologous strains of virus? If so, were they different?

Line 217 Methods; End point titres are not ideal as a measure of antibody response, as they can be subjective. It would be preferable to quantify the IgG response by constructing a standard curve for mouse IgG and measuring your responses against that. When you state that the endpoint titre was the reciprocal of that last dilution which gave  a positive response, please clarify that that was after the subtraction of your background reading on the ELISA plate?

 Figure 4: there appear to be spleen samples from only 3 animals in the control groups versus 6 in the treatment groups? 

In Figure 4, why don't you compare the groups with each other, rather than compare all responses against the group that received LNP's with no mRNA?

Lines 477-8; you state that 30ug mRNA in LNP's caused adverse effects in the mice, thus the dose was reduced to 10ug. Were there any adverse effects when you delivered 30ugmRNA by Jet injection?

figure3: the 'intact' group is more usually referred to as 'naive'

Author Response

We would like to thank the reviewer for careful and thorough reading of our manuscript and for the thoughtful comments and constructive suggestions, which help to improve its quality. Our response follows.

Corrections to the text of MS are highlighted in green.

[Comments 1]: Line 29: Protectivity is more usually 'protection' or 'efficacy'.

[Response 1]: The correction has been made.

[Comments 2]: Lines 191-200 Methods: can the authors say why they chose different dose-levels of mRNA for the 3 key immunisation groups? did they do a previous dose ranging study?

[Response 2]: We added the information to “Materials and Methods” at line 219-221:

«Consistent with the observed adverse effects of the high dose of LNPs in the first stage of the experiment, the dose of mRNA-H5-LNP in the second stage was reduced to 10 μg.»

At the initial and secondary stages of the experiment, Group 3 comprised of animals immunized with a recombinant HA/H5 protein, the immunogenicity of which had been previously documented and used as a positive control. The dosage of 25 μg of protein was determined through experimental means in accordance with data that had been previously obtained.

[Comments 3]: Lines 262: Did the authors calculate he MLD50 for both the homologous and heterologous strains of virus? If so, were they different?

[Response 3]: We added the information to “Materials and Methods” at line 304-307:

«Mice were infected intranasally with 20 MLD50 (MLD50 = 4.4±0.4 lg EID50 (embryonic infectious dose)) of the influenza virus strain A/Astrakhan/3212/2020 (H5N8) or 20 MLD50 (MLD50 = 2.0±0.4 lg EID50) of the influenza virus strain A/chicken/Magadan/14-7V/2022 (H5N1)».

[Comments 4]: Line 217 Methods; End point titres are not ideal as a measure of antibody response, as they can be subjective. It would be preferable to quantify the IgG response by constructing a standard curve for mouse IgG and measuring your responses against that. When you state that the endpoint titre was the reciprocal of that last dilution which gave a positive response, please clarify that that was after the subtraction of your background reading on the ELISA plate?

[Response 4]: We agree that the endpoint titers are not ideal for determining the level of humoral immune response because of differences between laboratories. However, when it is not possible to use a standardized commercial test system, researchers are forced to develop their own ELISA system using purified recombinant protein as an antigen and present the results as end point titers.
In this experiment, it was important to show that the sera of animals in the experimental groups contain antibodies that specifically bind to HA/H5, and to compare the ELISA results of animal sera from the experimental groups with the positive control group and non-immunized animals. To quantify the IgG response by constructing a standard curve for mouse IgG, we need to have a standardized set. We believe that these data would not greatly affect the overall picture of the humoral immune response.

We added the information to “Materials and Methods”, section 2.9:

«The serum dilution at which the optical density value was more than twice that of the negative control (in which blocking buffer was added to the wells instead of serum) was the last dilution which gave a positive response.».

[Comments 5]: Figure 4: there appear to be spleen samples from only 3 animals in the control groups versus 6 in the treatment groups?

[Response 5]: In the control groups, samples from only three animals were examined. In the experimental groups, six animals each were examined. This is due to limited laboratory resources for this study. As for the validity of this approach, reducing the number of animals in the control group is acceptable when applying the criterion used in our study, and it does not affect the adequacy of the statistical evaluation of the experimental results.

[Comments 6]: In Figure 4, why don't you compare the groups with each other, rather than compare all responses against the group that received LNP's with no mRNA?

[Response 6]: Comparisons were made between all groups. Only the statistically significant differences are marked in the figure. The graph does not indicate the absence of statistical differences. We acknowledge that this could lead to misunderstandings, so the following clarification was added to Section 2.14 of the Materials and Methods:

«Comparisons were not statistically significant unless otherwise stated».

[Comments 7]: Lines 477-8; you state that 30ug mRNA in LNP's caused adverse effects in the mice, thus the dose was reduced to 10ug. Were there any adverse effects when you delivered 30ug mRNA by Jet injection?

[Response 7]: No adverse effects were observed in the group in which mRNA was injected using JI.

We added the information to “Discussion”:

«However, these phenomena were not observed in the mRNA-H5 JI group».

[Comments 8]: Figure3: the 'intact' group is more usually referred to as 'naive'

[Response 8]: The correction has been made.

Reviewer 3 Report

Comments and Suggestions for Authors
  • While both LNP and JI delivery demonstrated full protection in mice, a more detailed comparison of the immune response kinetics and durability between the two delivery routes is needed to understand long-term implications.
  • The higher humoral response observed with LNP delivery is clearly presented, but adverse events in that group are only briefly discussed. Please elaborate on the observed side effects and discuss their implications for safety and dose optimization.
  • The manuscript reports full protection against heterologous challenge (H5N1), which is notable. However, additional discussion or data on cross-neutralization or breadth of immune response would significantly strengthen the claim of broad protection.
  • It is stated that LNPs may cause adverse systemic effects, and this is partly why JI is proposed as an alternative. Please provide more quantitative data or references to support these safety concerns, and clarify whether any such issues were observed in this study.
  • The discussion of antigen dose suggests that increasing the mRNA dose does not proportionally improve the immune response. This is an important point—please support this with quantitative data or prior studies, and explain whether this finding is delivery-method specific.
  • The sample sizes in each group (e.g., n=12 or n=20) should be justified with a power calculation or explanation for statistical adequacy, particularly for the challenge experiments.
  • The statistical analysis is broadly appropriate, but figures would benefit from clearer labeling of significance, inclusion of p-values in the legends, and clarification of the number of replicates for each data point.
  • While mice are a standard model, please briefly discuss the limitations of murine models for evaluating influenza vaccines intended for broader use, and outline your plans for testing in ferrets or other relevant models.
  • The presentation of LNP characterization data (size, charge, encapsulation) is solid, but could be strengthened by including batch consistency information or error margins across replicates.
  • Please revise some parts of the manuscript for language clarity, especially in the Results and Discussion sections where certain points are repeated or overly detailed.

Author Response

We would like to thank the reviewer for careful and thorough reading of our manuscript and for the thoughtful comments and constructive suggestions, which help to improve its quality. Our response follows.

Corrections to the text of MS are highlighted in green.

[Comments 1]: While both LNP and JI delivery demonstrated full protection in mice, a more detailed comparison of the immune response kinetics and durability between the two delivery routes is needed to understand long-term implications.

[Response 1]: We agree that this aspect requires more detailed consideration. Prolonged circulation of LNP in the body may contribute to a longer-lasting immune response. As a continuation of this experiment, we have already planned to study the kinetics of the immune response and its stability over time. The aim of this study was only to evaluate the immunogenicity and protective properties of the vaccine obtained, as well as to evaluate the effectiveness of methods for delivering it to mice.

[Comments 2]: The higher humoral response observed with LNP delivery is clearly presented, but adverse events in that group are only briefly discussed. Please elaborate on the observed side effects and discuss their implications for safety and dose optimization.

[Response 2]: This study did not conduct a detailed investigation of the adverse effects of administering mRNA-LNPs to mice. Only the observed adverse effects documented in connection with a 30-μg dose of mRNA-LNP were considered. This aspect was highlighted to explain the reduction in dosage for the second phase of the study, in accordance with bioethical and humanity principles, to minimize harm to animals, as discussed at Lines 514-526

In subsequent studies, we intend to examine the dose-dependent effect of the vaccine in greater detail and concurrently investigate the adverse effects of our composition at various doses.

[Comments 3]: The manuscript reports full protection against heterologous challenge (H5N1), which is notable. However, additional discussion or data on cross-neutralization or breadth of immune response would significantly strengthen the claim of broad protection.

[Response 3]: We added the information to «Discussion»:

«One potential explanation for the complete protection of immunized mRNA-H5 mice against heterologous challenge is the presence of sufficient amino acid sequence similarity between the HA in strains of avian influenza virus heterologous to the A/turkey/Stavropol/320-01/2020 (H5N8) - A/Astrakhan/3212/2020 (H5N8) and A/chicken/Magadan/14-7V/2022 (H5N1). The degree of homology between the HA of the specified A/H5N8 and A/H5N1 strains was determined, revealing a high degree of similarity, with one amino acid substitution (V11I) in A/chicken/Magadan/14-7V/2022 (H5N1) in the head domain of hemagglutinin. This domain is re-sponsible for binding to host cell receptors and is a target for virus-neutralizing antibodies. The phenomenon of cross-protective immunity will be the subject of further investigation in forthcoming experiments that will utilize A/H5N8 and A/H5N1 strains from disparate geographic regions.

Additionally, our earlier study of a DNA vaccine encoding an immunogen similar to that in mRNA-H5 [43] yielded data on cross-reactivity against heterologous strains. The pVAX-H5 DNA vaccine was shown to provide 100% protection against the HPAI virus strain A/chicken/Khabarovsk/24-1V/2022 (H5N1) in mice. This strain has significant antigenic differences from the one on which the immunogen was based. This fact may indicate the immunogen's ability to provide broad protection against various virus strains»

[Comments 4]: It is stated that LNPs may cause adverse systemic effects, and this is partly why JI is proposed as an alternative. Please provide more quantitative data or references to support these safety concerns, and clarify whether any such issues were observed in this study.

[Response 4]:  In this MS, we discuss the adverse effects reported in the literature when using LNPs as a delivery method. To a significant extent, these phenomena are attributable to the pervasive systemic distribution, as referenced at Lines 66-68. This issue is further elaborated upon in the "Discussion" section, specifically at Lines 471-483.

In accordance with the recommendation, we are incorporating a number of references in the "Introduction" and "Discussion" sections to relevant studies on this subject in the literature.

Regarding the observation of these issues in our study, we did not investigate adverse effects, as that was not the objective of our study. In this work, we only briefly discuss the visible adverse effects in mice when using the dose of LNPs necessary to package 30 μg of mRNA. It is important to note that when naked mRNA was administered at a dose of 30 μg by JI, no adverse effects were observed in mice. It is evident that publications predominantly address the issue of adverse effects in clinical trials, as well as in the long term, a problem that is unfeasible in the context of exploratory research.

[Comments 5]: The discussion of antigen dose suggests that increasing the mRNA dose does not proportionally improve the immune response. This is an important point—please support this with quantitative data or prior studies, and explain whether this finding is delivery-method specific.

[Response 5]:  Indeed, we observed that antibody titers exhibited slight variations when different doses of mRNA (10 and 30 μg) were used, as demonstrated in Figures 3 and 5. In the subsequent "Discussion" section, this aspect is examined from the perspectives of both the specificity of the delivery method and the potential of the mRNA itself.

We supplement the discussion with references to additional studies that have encountered similar results, as well as reasons for this phenomenon.

[Comments 6]: The sample sizes in each group (e.g., n=12 or n=20) should be justified with a power calculation or explanation for statistical adequacy, particularly for the challenge experiments.

[Response 6]:  The number of animals allocated to each group was determined by ethical considerations and the necessity to ensure statistical significance. The sample size of n=20 animals was determined by the experimental design, which entailed investigating cross-immunity following infection with two distinct virus strains. In this stage of study, each group was divided into two subgroups of 10 animals, and subsequently each subgroup was infected with either strain A/Astrakhan/3212/2020 (H5N8) or strain A/chicken/Magadan/14-7V/2022 (H5N1). The number of animals was determined to be n=10, as it was expected that some of the animals would perish from the infection with the virus.

The design of the second stage of experiments was described in more detail in «Materials and Methods»:

«The animals were divided into 8 groups of 10 animals each and immunized on days 0 and 21: Group 1 and 2 was immunized with 10 μg of mRNA-H5-LNP; Group 3 and 4 was immunized with 30 μg of mRNA-H5 using JI; Group 5 and 6 was immunized with 25 μg of recombinant HA/H5 protein with aluminium hydroxide; Group 7 and 8 consisted of non-immunized animals».

[Comments 7]: The statistical analysis is broadly appropriate, but figures would benefit from clearer labeling of significance, inclusion of p-values in the legends, and clarification of the number of replicates for each data point.

[Response 7]:  We updated figures.

[Comments 8]: While mice are a standard model, please briefly discuss the limitations of murine models for evaluating influenza vaccines intended for broader use, and outline your plans for testing in ferrets or other relevant models.

[Response 8]:  We added the information to «Discussion»:

«However, it is important to acknowledge that although mice are the most commonly used animal model for the preliminary assessment of influenza vaccines, they do not naturally carry the virus, in contrast to humans. The predictive value of results obtained in mice may be affected by differences in disease manifestations, immune response, and virus transmission characteristics when evaluating vaccine efficacy in humans. Consequently, the subsequent phase of our research entails the administration of the vaccine to ferrets, which are regarded as a more relevant model».

[Comments 9]: The presentation of LNP characterization data (size, charge, encapsulation) is solid, but could be strengthened by including batch consistency information or error margins across replicates.

[Response 9]:  We added the information to «Results» in section 3.2:

«Characterization data for formulations obtained for subsequent immunizations are not provided, since the deviation from the presented data for each of the subsequent formulations was less than 10% according to DLS and ELS data».

[Comments 10]: Please revise some parts of the manuscript for language clarity, especially in the Results and Discussion sections where certain points are repeated or overly detailed.

[Response 10]:  Agree. We have, accordingly, revised «Results» and «Discussions» on the language clarity.

Round 2

Reviewer 3 Report

Comments and Suggestions for Authors
  • Please provide supporting data or references regarding the expected durability of immune responses with LNP versus jet injection delivery.
  • Adverse effects observed in the 30 μg LNP group are described qualitatively. Quantitative scoring or photographic evidence should be included to substantiate these observations.
  • The safety concerns related to LNP systemic effects are literature-based. This limitation should be clearly acknowledged in the Abstract and Discussion.
  • Cross-protection against H5N1 is attributed to HA homology, but no cross-neutralization data are shown. Please clarify this limitation explicitly.
  • Include statistical comparisons (e.g., p-values) between 10 μg and 30 μg dose groups to support the conclusion of a dose–response plateau.
  • A power calculation or explicit justification of sample sizes used in survival studies should be included.
  • Batch-to-batch variation of LNPs is stated as <10%, but supporting data are not shown. Please include this in a supplementary table.
  • Clarify in figure legends the number of biological replicates and whether shown values represent means with standard error or full range.
  • Please provide a brief outline or timeline for the planned ferret model study to support the translational trajectory.
  • Some discussion points—especially regarding dose effects and LNP drawbacks—are repetitive. Streamlining the text would improve clarity.

Author Response

We would like to thank the reviewer for careful and thorough reading of our manuscript and for the thoughtful comments and constructive suggestions, which help to improve its quality. Our response follows.

Corrections to the text of MS are highlighted in green.

[Comments 1]: Please provide supporting data or references regarding the expected durability of immune responses with LNP versus jet injection delivery.

[Response 1]:

We added the information to «Discussion»:

«Furthermore, a study by Abbasi et al. [27] demonstrated that JI of naked mRNA elicited a robust immune response in mice and non-human primates, with spike-specific IgG levels persisting at elevated levels for over six months following booster therapy. This finding indicates the presence of persistent humoral immune responses after JI of naked mRNA. The authors observe that the two approaches, LNPs and JI, result in comparable humoral immune responses with minimal systemic inflammation, suggesting the progressive effectiveness of these methods in immune response stimulation, despite differences in delivery mechanisms. In addition, Wang et al. [28] evaluated the safety and immune response to the RBD3-Fc vaccine administered as naked mRNA using JI compared to mRNA-LNP. The prime-boost immunization protocol elicited a robust immune response, and intradermal administration demonstrated comparable or even superior outcomes in antibody binding, neutralizing antibodies, and T-cell response. The efficacy of naked mRNA vaccines was found to be inferior to that of mRNA-LNP vaccines. Moreover, the presence of long-term antibody persistence was detected four weeks after booster therapy, suggesting the formation of memory B cells. The study demonstrated that rats vaccinated with mRNA-LNP using JI elicited robust systemic immune responses, elevated virus-neutralizing antibody titers, and substantial Th1 and Th2 immune responses. However, a direct comparison of the duration of the immune response between mRNA-LNP and JI of naked mRNA under identical conditions was not performed. Consequently, it can be concluded that both methods of mRNA vaccine delivery demonstrate efficacy at therapeutic doses, thereby opening up opportunities for further research into the relative durability of immune responses».

[Comments 2]: Adverse effects observed in the 30μg LNP group are described qualitatively. Quantitative scoring or photographic evidence should be included to substantiate these observations.

[Response 2]: We agree that quantitative determination would be more illustrative when considering the negative effects of LNPs. However, in this study, adverse effects were not the primary focus of the research. They were only incidentally observed during high-dose administration and, as a result, were not measured or documented. We do not focus on this fact in detail but mention it in accordance with the need for the ethical treatment of animals. In subsequent studies, we will prioritize investigating the negative effects and plan to do so in greater detail, as noted in our plan for further work.

We added the information to «Discussion»:

«These adverse effects have not yet been studied in detail, but will be examined in future studies».

[Comments 3]: The safety concerns related to LNP systemic effects are literature-based. This limitation should be clearly acknowledged in the Abstract and Discussion.

[Response 3]:

We added the information to «Discussion»:

«…adverse effects discussed in our current study based only on literature sources».

We added the information to «Abstract»:

«… and according to the literature, JI is safer than delivery using LNP».

[Comments 4]: Cross-protection against H5N1 is attributed to HA homology, but no cross-neutralization data are shown. Please clarify this limitation explicitly.

[Response 4]:

We added the information to «Materials and Methods»:

«The antigenic properties of viruses (based on hemagglutination-inhibition (HI) test and microneutralization) are presented in the table (Table 1).

Table 1. Antigenic properties of viruses used in study. »

Viruses

Titer with аnti-A/Astrakhan/3212/2020

ferret reference serum

HI-test

Microneutralization

A/Astrakhan/3212/2020 H5N8

640

320

A/turkey/Stavropol/320-01/2020 H5N8 2.3.4.4b

320

80

A/chicken/Magadan/14-7V/2022 H5N1 2.3.4.4b

160

80

We added the information to«Discussion»:

«As illustrated in Table 1, the antigenic properties of the A/chicken/Magadan/14-7V/2022 (H5N1) and A/Astrakhan/3212/2020 (H5N8) viruses exhibit minimal variation. The titers of reference serum with these viruses in the HI-test and in microneutralization differ by no more than fourfold. Therefore, the cross-protection against H5N1 is attributed to HA homology».

[Comments 5]: Include statistical comparisons (e.g., p-values) between 10μg and 30μg dose groups to support the conclusion of a dose–response plateau.

[Response 5]:

Indeed, the aforementioned data would likely prove to be highly indicative. However, a statistical comparison of the indicators of these two groups is not possible, as the animals were obtained from different experiments. Combining and comparing them would not be indicative. The conclusion is derived from a comparative analysis of the experimental groups that received varying doses of the mRNA-H5-LNP with the group that remained unchanged in both experiments (the group to which mRNA-H5 was administered using JI). Statistically significant differences (p < 0.05) between the immune response indicators of these groups in both the first and second experiments indicate that no significant differences in the immune response were found.

[Comments 6]: A power calculation or explicit justification of sample sizes used in survival studies should be included.

[Response 6]:

In our previous studies of the pVAX-H5 DNA vaccine, which encodes a similar mRNA-H5 immunogen, we used G*Power 3.1 software to calculate the expected power of the criterion for determining the sufficient sample size to study protective efficacy. The studies can be found here: https://doi.org/10.3390/vaccines12050538 and https://doi.org/10.3390/vaccines17030330. Wilcoxon–Mann–Whitney (two groups), effect size = 1.44, and desired power (1-β) = 0.8. This was based on data from the variability of the "animal body weight" indicator when the animals were infected with the A/Astrakhan/3212/2020 (H5N8) and A/chicken/Khabarovsk/24-1V/2022 (H5N1) viruses. Because the "animal body weight" indicator can be highly variable (from 0.36 to 2.03 g), we used groups of ten animals per infection to achieve a statistically significant difference.

Upon analyzing the results obtained for mRNA-H5, we observed 100% survival in the experimental groups. Using the Wilcoxon–Mann–Whitney statistical test, we found significant differences in animal weight after infection between the mRNA-H5-LNP group (p = 0.0002) and the mRNA-H5-JI group (p < 0.0001) compared to the control group at α = 0.05.

Therefore, the sample size we have determined (n=10) is sufficient to obtain statistically reliable data.

We added the information to «Materials and Methods»:

«The number of animals in each group was determined based on the findings of prior studies of a similar immunogen in the form of a DNA vaccine [36,39]».

[Comments 7]: Batch-to-batch variation of LNPs is stated as <10%, but supporting data are not shown. Please include this in a supplementary table.

[Response 7]:We added the information to «Appendix»

Formulations

Average particle size, nm

Polydispersity index

ζ - potential

mRNA-H5-LNP (30 µg) (first immunization)

93.5±0.8

0.149 ± 0.01

-0.02±0.26

mRNA-H5-LNP (30 µg) (second immunization)

101.3±1.1

0.135 ± 0.03

-0.1±0.35

mRNA-H5-LNP (10 µg) (first immunization)

97.61±1.5

0.204±0.014

-0.037±0.01

mRNA-H5-LNP (10 µg) (second immunization)

100.7±0.4

0.214±0.032

-0.042±0.062

[Comments 8]: Clarify in figure legends the number of biological replicates and whether shown values represent means with standard error or full range.

[Response 8]: We added information to the figure legends.

[Comments 9]: Please provide a brief outline or timeline for the planned ferret model study to support the translational trajectory.

[Response 9]:

We added the information to «Discussion»:

«Within a year, we plan to conduct mRNA-H5 trials on ferrets, which are a more relevant model. Prior to this, a comprehensive analysis of the vaccine is planned, which will be conducted on smaller animal models (e.g., BALB/c mice). This analysis will include a more precise determination of the optimal dose for each vaccine variant. Immunization of ferrets with mRNA vaccine in LNP and with JI, and with recombinant HA/H5 protein is planned. The immune response will be evaluated using ELISA, microneutralization, and ELISpot. The ability of mRNA-H5 to protect animals (ferrets) from infection with HPAI viruses A/H5N8 and A/H5N1 strains from disparate geographic regions will also be investigated. In addition, the following procedures will be carried out: quantification of viral RNA in ferret organ samples, pathological analyses of lung tissue, and assessment of influenza virus reproduction in the lungs. The selection of the model is predicated on the potential for observing clinical manifestations or the absence thereof, similar to those observed in humans, that correspond to the influenza. Furthermore, we plan to study the adverse effects discussed in our current study based only on literature sources. The study will include an assessment of toxicity, allergenic properties, mutagenicity, histopathological changes, and a number of other indicators using guinea pigs and/or rabbits as model animals».

[Comments 10]: Some discussion points—especially regarding dose effects and LNP drawbacks—are repetitive. Streamlining the text would improve clarity.

[Response 10]:

The correction has been made. Lines 64-66 and 499-500 were deleted to avoid repetition. In accordance with Reviewer 2's recommendations, information about doses and effects, which is repeated in "Materials and Methods" and then disclosed in "Discussions," has been added. The detailed disclosure of this issue and the issue of LNP shortcomings has been made in accordance with the recommendations from the first round of reviews.